# Comparative transcriptome analysis reveals key genes potentially related to organic acid and sugar accumulation in loquat

Jun Yang[1], Jing Zhang[1], Xian-Qian Niu[2], Xue-Lian Zheng[1], Xu Chen[1], Guo-Hua Zheng[1]*, Jin-Cheng Wu[3]*

1 College of Horticulture, Fujian Agriculture and Forestry University, Fuzhou, PR China, 2 Fujian Science Technology of Tropical Crops, Zhangzhou, Fujian, China, 3 College of Environmental and Biological Engineering, Putian University, Putian, China

* fafuzgh@126.com (GHZ); wjc2384@163.com (JCW)

**Data Availability Statement:** Data is available via SRA accession number PRJNA579044 (https://www.ncbi.nlm.nih.gov/search/all/?term=

## Abstract

Organic acids and sugars are the primary components that determine the quality and flavor of loquat fruits. In the present study, major organic acids, sugar content, enzyme activities, and the expression of related genes were analyzed during fruit development in two loquat cultivars, 'JieFangZhong' (JFZ) and 'BaiLi' (BL). Our results showed that the sugar content increased during fruit development in the two cultivars; however, the organic acid content dramatically decreased in the later stages of fruit development. The differences in organic acid and sugar content between the two cultivars primarily occured in the late stage of fruit development and the related enzymes showed dynamic changes in activies during development. Phosphoenolpyruvate carboxylase (PEPC) and mNAD malic dehydrogenase (mNAD-MDH) showed higher activities in JFZ at 95 days after flowering (DAF) than in BL. However, NADP-dependent malic enzyme (NADP-ME) activity was the lowest at 95 DAF in both JFZ and BL with BL showing higher activity compared with JFZ. At 125 DAF, the activity of fructokinase (FRK) was significantly higher in JFZ than in BL. The activity of sucrose synthase (SUSY) in the sucrose cleavage direction (SS-C) was low at early stages of fruit development and increased at 125 DAF. SS-C activity was higher in JFZ than in BL. vAI and sucrose phosphate synthase (SPS) activities were similar in the two both cultivars and increased with fruit development. RNA-sequencing was performed to determine the candidate genes for organic acid and sugar metabolism. Our results showed that the differentially expressed genes (DEGs) with the greated fold changes in the later stages of fruit development between the two cultivars were phosphoenolpyruvate carboxylase 2 (*PEPC2*), mNAD-malate dehydrogenase (*mNAD-MDH*), cytosolic NADP-ME (*cyNADP-ME2*), aluminum-activated malate transporter (*ALMT9*), subunit A of vacuolar H$^+$-ATPase (*VHA-A*), vacuolar H$^+$-PPase (*VHP1*), NAD-sorbitol dehydrogenase (*NAD-SDH*), fructokinase (*FK*), sucrose synthase in sucrose cleavage (*SS-C*), sucrose-phosphate synthase 1 (*SPS1*), neutral invertase (NI), and vacuolar acid invertase (*vAI*). The expression of 12 key DEGs was validated by quantitative reverese transcription PCR (RT-qPCR). Our findings will help understand the molecular mechanism of organic acid and sugar formation in loquat, which will aid in breeding high-quality loquat cultivars.

PRJNA579044) and via Dryad (https://datadryad.
org/stash/dataset/doi:10.5061/dryad.c866t1g5c).

**Funding:** Specific grant numbers: KFA18081A
Initials of authors who received each award:
Guohua Zheng Full names of commercial
companies that funded the study or authors:
Screening and analysis of key genes for malate
metabolism in loquat fruit Initials of authors who
received salary or other funding from commercial
companies:No URLs to sponsors' websites: No
Also state whether any sponsors or funders (other
than the named authors) played any role in: The
funders had no role in study design, data collection
and analysis, decision to publish, or preparation of
the manuscript.

**Competing interests:** I have declared that no
competing interests exist.

## 1. Introduction

Loquat (*Eriobotrya japonica* Lindl.), an evergreen fruit tree from the Rosaceae family, is indigenous to southeastern China. It is widely cultivated in many countries such as Japan, India, Israel, Brazil, Italy, and Spain. Loquats are harvested in an unusual season (from late spring to early summer) and thus, have an advantage over other fruits in the fresh fruit market [1]. The taste of loquats greatly contributes to their quality and governs their sales. Fruit taste primarily influences human selection, which is a major force in the selction of agricultural crops. Acids are one of the three major components of fleshy fruit taste, together with sugars and volatile flavor compounds [2]. The ratio of sugars to organic acids is critical to the taste trait [3, 4]; therefore, the mechanism of sugar and organic acid accumulation in loquat fruits has attracted much attention from food experts and researchers. A previous study identified a family of plant-specific genes that affect fruit acidity [5] in melons, which control fruit acidity across polant families.

Sugars and organic acids are the major soluble components of fruits and play an important role in the fruit taste and flavor. Fructose, sucrose, glucose, and sorbitol are the main sugars in loquat fruits and rapidly accumulate in the fruits during ripening [6]. Organic acids also contribute to the flavor and taste of loquat fruits and their content and ratio may vary among various cultivars [7]. Among the organic acids, malic acid is predominant in unripe loquats; however, it decreases as the fruit ripens [6]. The malate concentration in ripe fruit was determined by the balance of malate biosynthesis, degradation, and vacuolar storage [8]. Chen et al. [9] reported that malate and quinate were the major organic acids in both Jiefangzhong and Changhong 3, and malic acid biosynthesis and degradation could be regulated by phosphoenolpyruvate carboxylase (PEPC), NAD-malate dehydrogenase (NAD-MDH), and NADP-malic enzyme (NADP-ME) in loquat pulp. Ethychlozate treament of highly acidic 'JieFangZhong' (JFZ) pulp could weaken PEPC and NAD-MDH activities and enhance the activities of cytosolic NADP-ME (cyNADP-ME), thus reducing the content of malic acid in the pulp of loquat [10]. Moreover, *PEPC*, *NADP-ME*, *cyNAD-MDH*, *mNAD-MDH*, *V-ATPase A*, and *V-PPase* were first isolated from the loquat pulp [11]. As the organic acid content decreases during ripening, the sugar content increases, because organic acids may serve as a source of carbon for sugar production [12]. In a study on 12 cultivars of loquat fruits, soluble sugars were found to comprise fructose, sucrose, glucose, and sorbitol [13]. Fructose and sucrose contribute more to the sweet taste of loquat fruits than glucose [14]. Song et al. [15] reported that high expression levels of *EjSS-S* and *EjSPS1* could contribute to increased sucrose synthesis at stage I while *EjSS-C* primarily caused sucrose cleavage rather than sucrose synthesis from 100 to 150 post-anthesis days.

In recent years, advances in transcriptome sequencing technology have fostered comparative transcriptome analyses of sugar and organic acid metabolism during fruit development in many members of Rosaceae, including pear [16], plum [17], strawberry [18], Chinese bayberry [19], and sweet cherry [20]. In the present study, RNA-sequencing was performed to screen for key genes regulating sugar and organic acid metabolisms during the development of loquat fruit. Moreover, organic acid content, soluble sugar content, and enzyme activity were determined to study their relationships to gene expression levels during the development of JFZ and 'BaiLi' (BL) to gain insight into the mechanisms of organic acid and sugar accumulation during fruit development.

## 2. Materials and methods

### 2.1. Collection of materials

Loquat fruits from cultivars 'JieFangZhong' (JFZ) and 'BaiLi' (BL) grown in the orchards of Changtai County, Putian city, Fujian Province, China, were used as the test materials in this

study. Both cultivars are late maturing, and have similar flowering patterns. Thirty disease-free fruits of uniform size from each cultivar were collected at three developmental stages (65, 95, and 125 DAF). 10 pieces of fruits for each stage (cultivar) were collected to dtermination of sugar and acid content. The peel, pulp, and seeds of each sample were carefully separated. The pulp was wrapped in foil and immediately frozen in flash nitrogen and stored at −80˚C for subsequent use.

## 2.2. Determination of total soluble solids and total acidity

TSS was measured using a digital hand-held refractometer (PR101-α, Otago, Japan). Thirty fruits were harvested and randomly divided into three groups. The fruits in each group were juiced and the pooled juice was considered one biological replicate. Then, 5 mL of this pooled juice from each group was diluted with 20 mL distilled water. The total acidity (TA) of the pulp was determined by titration with 0.1 N NaOH to the end point at pH 8.1 and the TA was expressed as malate; this was performed according to Chen et al. [21].

## 2.3. Extraction and determination of organic acids, soluble sugars, and starch

Organic acids and sugars were extracted by a method described by Chen et al. [21]. One gram of frozen pulp sample was ground in liquid nitrogen using a chilled mortar and pestle. The powder was dissolved in 3 mL of 80% (v/v) ice-cold methanol. Then, the extraction was ultra-sonicated (Fisher Scientific FS 60, UK) for 30 min at 50˚C, and centrifuged at 8000 g for 10 min. The sample was repeatedly extracted three times, and the precipitate was used to measure starch. The precipitate was digested with 2 M KOH and the supernatant was used to test the content of glucose. Starch was expressed as glucose equivalents [22]. The extracts were analyzed using a Waters HCLASS system equipped with PDAeλ detector. A 10 μl sample was injected into a C18 column (2.1 mm × 100 mm, 1.8 μm, Waters Technologies, WA, USA) to determine the malic acid content. The mobile phase was methanol: 0.01 mol/L $K_2HPO_4$ solution (pH = 2.5) with a flow rate of 0.2 mL/min. The column temperature was set at 25˚C. The detection wavelength was 210 nm. The chromatographic conditions for soluble sugars were acetonitrile: deionized water (80:20) with a flow rate of 1 mL min$^{-1}$. A 5.0 μm NH2 (4.6 mm × 250 mm) column and refractive index detector RI-2414 (Waters Technologies) was used. Organic acid and sugar contents were calculated by a standard curve method and expressed as mg g$^{-1}$ FW of fruit. There were three replicates for the measurement of organic acids and sugars. All organic acid and sugar standards were obtained from Sigma–Aldrich (Sigma, WA, USA).

## 2.4. Extraction and determination of organic acids and sugars metabolism related enzyme activity

The activities of PEPC, NAD-MDH, and NADP-ME were extracted according to Chen et al. [9]. All enzyme activities were monitored at 340 nm. PEPC activity was measured in 1 mL reaction mixture containing 50 mM Tris-HCl (pH 8.0), 5 mM $MgCl_2$, 2 mM dithiothreitol (DTT), 10 mM NaHCO3, 0.2 mM NADH, 5 units of NAD-MDH, 2.5 mM phosphoenolpyruvate (PEP), and 30 μl of extract. The reaction was initiated by 2.5 mM PEP. NAD-MDH activity was assayed in a 1 mL reaction mixture containing 50 mM Tris-HCl (pH 7.8), 2 mM $MgCl_2$, 0.5 mM EDTA, 0.2 mM NADH, 2 mM oxaloacetate, and 30 μl of extract. The reaction was initiated by 2 mM oxaloacetate. NADP-ME activity was determined in a 1 mL reaction mixture containing 80 mM Tris–HCl (pH 7.5), 0.1 mM EDTA, 1 mM DTT, 0.2 mM NADP,

0.4 mM MnSO4, 3 mM malate, and 100 μl of extract; the reaction was initiated with 3 mM malate.

SDH, AI, and SPS activities were determined according to Li et al. [23], with some modifications. SDH was assayed in a reaction mixture (1 mL) consisting of 300 mM Sor, 1 mM NAD$^+$, and 0.2 mL of the desalted extract in 100 mM Tris-HCl (pH 9.6), and NADH production was determined at 340 nm. vAI was determined in a 200 mL assay mixture containing 100 mM phosphate-citrate buffer (pH 4.8), 100 mM sucrose, and 50 mL of the desalted extract or denatured extract (as blank). The assays were stopped by boiling for 3 min before adding 0.75 M Tris-HCl buffer (pH 8.5). SPS was measured in a reaction mixture (200 μl total volumes) containing 50 mM HEPES-KOH (pH 7.4), 4 mM MgCl$_2$, 1 mM EDTA, 4 mM F6P, 20 mM G6P, 3 mM UDPG, and 100 mL of sample. The mixture was incubated at 27˚C for 30 min, and then boiled for 3 min to stop the reaction. FRK was assayed according to the methods described by Qin et al. [24]. SS activity in the direction of sucrose degradation (SS-C) was measured in a solution containing 20 mM HEPES-NaOH (pH 7.8), 100 mM sucrose, and 5 mM UDP. Incubation was carried out at 25˚C for 30 min and then the reaction stopped by heating at 95˚C for 5 min. The determination of UDPG was performed in 200 mM glycine (pH 8.9), 5 mM MgC12, and 0.6 mM NAD, and the reaction was started with 0.05 U UDPG dehydrogenase [25].

## 2.5. Total RNA extraction, cDNA library construction and sequencing

Total RNA from the fruit pulp of BL and JFZ for each fruit stage was isolated using plant RNA Midi Kit (OMEGA, Guangzhou, China) according to the manufacturers instructions. Three biological replicates were taken to extract total RNA. The quality of RNA samples was determined using NanoDrop 2000 (Thermo Fisher Scientific Inc, Carlsbad, USA). The integrity of the total RNA was checked using the Agilent 2100 Bioanalyzer. RNA-Seq libraries were constructed using the protocol described in Zhong et al. [26] and sequenced on Illumina HiSeq2000 platform in the paired-end mode.

## 2.6. De novo assembly and transcript annotations

Raw data (raw reads) were filtered using FastQC software [27]. The adapter sequences, reads containing more than 10% of unknown nucleotides (N), and low-quality reads containing more than 40% of low quality (Q-value ≤ 10) bases were removed for a more accurate assembly of the transcriptome data. The Q20, GC content, and sequence duplication level of the clean data were calculated. All clean reads were assembled using a short-read assembling program, Trinity (v2.4.0) [28], with the minimum kmer coverage set to 10. The cleaned reads were then aligned to the assembled contigs using Bowtie [29]. We used the blastx program with an E-value threshold of 1e-5 to align unigenes sequences to NCBI non-redundant protein database (Nr), the Swiss-Prot protein database (Swissprot), the Kyoto Encyclopedia of Genes and Genomes database (KEGG), and Cluster of Orthologous Groups of proteins (COG/KOG), and then obtained the protein function annotation information of the unigenes.

## 2.7. Identification of DEGs related to organic acids and sugars metabolism

To calculate the unigene expression, we used the fragments per kilobase of transcript per million mapped reads (FPKM) method. Differentially expressed genes across samples or groups were identified by using the edgeR [30]. We tested genes with a fold change ≥ 2 and a false discovery rate (FDR) < 0.05 in comparison as DEGs. The DEGs involved in organic acid and sugar metabolisms were searched based on candidate gene annotation information, then sequences identified were further confirmed by the NCBI BLAST website (https://blast.ncbi.

nlm.nih.gov). The RNA-Seq data were deposited in the NCBI database (https://www.ncbi.nlm.nih.gov/sra/) with an SRA accession number SUB6454097.

## 2.8. RT-qPCR analysis

All primer pairs were designed using the Beacon Designer 7.0 software, and primer sequences are shown in S3 Table. Total purified RNAs were reverse transcribed to cDNAs using a Prime-Script Master Mix Kit (Takara, Fujian, China) following the manufacturers instructions. Real-time PCR systems and procedures are performed using the SYBR$^{\circledR}$ Premix Ex Tap$^{TM\ II}$ kit (Takara, Japan) on a qTOWER2.2 instrument (Jena, Germany). RT-qPCR was performed under the following conditions: 95°C for 30 s, 95°C for 5 s, and 45 cycles of 60°C for 20 s, and a melt curves program (60–95°C). Data were calculated according to the $2^{-\Delta\Delta CT}$ method. The transcript levels were normalized relative to the actin gene. Three biological and three technical replicates were used for the RT-qPCR assays.

## 2.9. Statistical analysis

Results are expressed here as mean ± SD. The differences among means were evaluated using Tukeys multiple comparison test and the statistical significance was set at $p < 0.05$. Verification of transcriptome data was carried out using R (corrplot package). Significant correlation between data groups is indicated if p-value is less than 0.05 ($p < 0.05$).

# 3. Results

## 3.1. Changes in organic acid and sugar content during fruit development

We measured fruit weight and content of organic acids and sugars at three different fruity ripening stages: unripe (65 days after flowering (DAF)), maturity (95 DAF), and over-ripening (125 DAF). The fresh weight in both cultivars increased during fruit development. The fresh fruit weight of JFZ was significantly higher than BL at the maturity and over-ripening stages (Fig 1A). Tartaric acid (TA) in the two cultivars peaked at 95 DAF (Fig 1B). As shown in Fig 1C, in both cultivars, the starch content peaked at 95 DAF, and suddenly decreased thereafter. The two cultivars did not show significant differences in glucose content (Fig 1D). JFZ showed a higher starch content compared with BL at 125 DAF. BL had a significantly higher total soluble solid (TSS) content than JFZ at maturity (Fig 1E). Malic acid (10–16 mg/g fresh weight (FW)) was the dominant organic acid at maturity and showed a similar trend to TA in both cultivars during fruit development (Fig 1F). Malic acid content was significantly higher in JFZ than in BL at 125 DAF. In mature JFZ and BL fruits, fructose was the main soluble sugar. Fructose content increased steadily with fruit development from 65 to 125 DAF, and it was significantly higher in BL than in JFZ at 125 DAF (Fig 1G). Sucrose content in both cultivars peaked at 95 DAF and decreased thereafter. The sucrose content in JFZ was significantly higher than in BL at 125 DAF (Fig 1H).

## 3.2. Transcriptome sequencing, de novo assembly and functional annotation analysis

A total of 18 cDNA libraries were obtained from JFZ and BL fresh fruits at the three critical ripening stages (with three biological replicates for each stage and cultivar), and then sequenced on Illumina HiSeq$^{TM}$ 2000 platform. Approximately 50 million raw reads were produced for each library. After removing the low quality and adapter reads, clean reads within each library were retrieved, with quality scores at the Q20 level (Table 1). The clean reads were assembled using the Trinity method, yielded 45,041 unigenes (Table 2). We submitted the

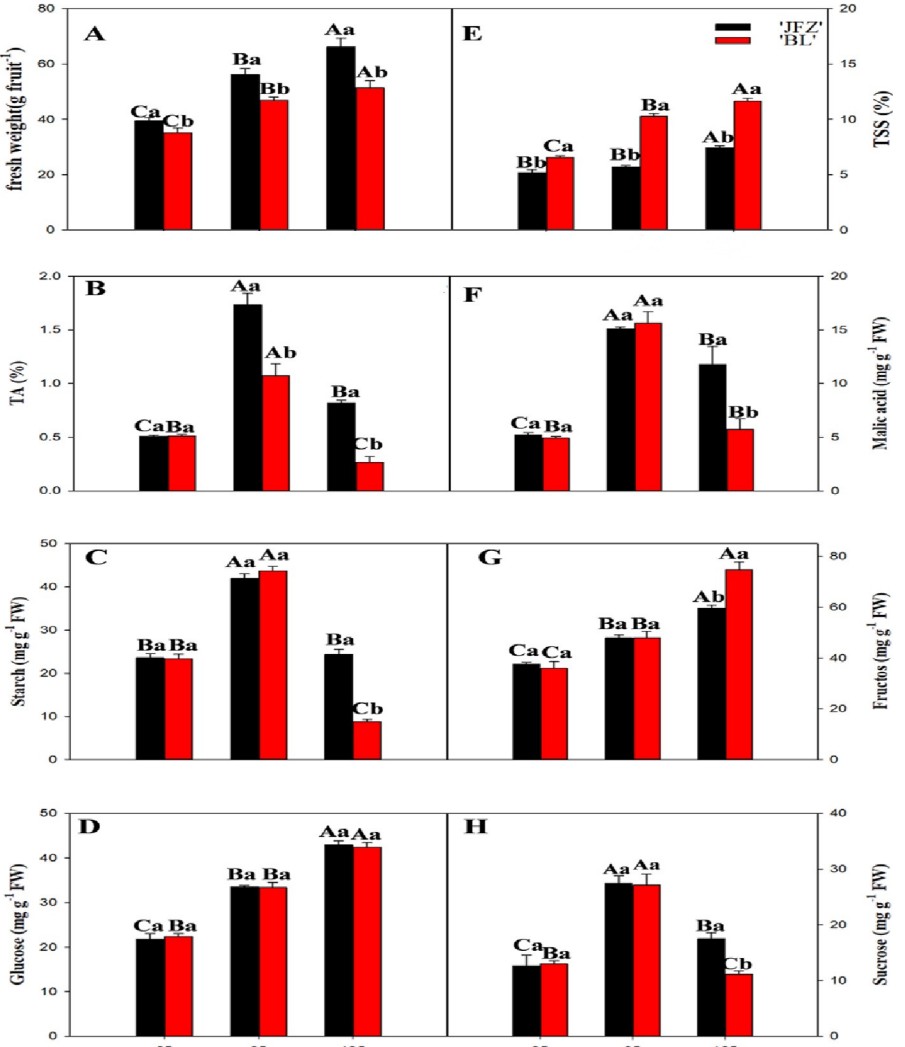

**Fig 1. Changes in average fresh weight and content of fresh juice total soluble solids, total acid, malic acid, starch, fructose, glucose, and sucrose in the cultivars JFZ and BL loquat fruits during development and ripening.** Differences among the samples were analyzed by two (cultivars) × three (sampling dates) ANOVA. The different lowercase letters indicate significant differences between the two cultivars at the same time point at $p < 0.05$. The different capital letters indicate significant differences among different time points for the same cultivar at $p < 0.05$. Each point represents mean ± SD (n = 3).

unigenes to four databases, revealing that 32,533 (72.22%), 21,778 (48.34%), 18,119 (40.23%) and 12,067 (26.79%) unigenes were annotated in the Non-redundancy Protein database (NR), the Swiss-Prot database (Swiss Prot), the Kyoto Encyclopedia of Genes and Genomes database (KEGG), and the Cluster of Orthologous Groups of proteins database (COG), respectively (Fig 2).

### 3.3. Analysis of Differentially Expressed Genes (DEGs)

DEGs between the two cultivars were identified by using FDR ≤ 0.05 and $\log_2$ fold change ≥ 1 as the thresholds. DEGs detected are presented in Fig 3A. The relationships between different DEG groups are shown as Venn diagrams (Fig 3B). Because the differences in sugar and

**Table 1. Summary of the sequencing data analyses.**

| Samples | Raw Reads | Clean Reads (%) | GC (%) | Low Quality (%) | Q20 (%) |
|---|---|---|---|---|---|
| J65-1 | 51166572 | 49768850 (97.27%) | 49.3 | 1113480 (2.18%) | 98.01 |
| J65-2 | 53886878 | 52336300 (97.12%) | 49.64 | 1252522 (2.32%) | 97.91 |
| J65-3 | 56245656 | 54700842 (97.25%) | 49.08 | 1233374 (2.19%) | 97.99 |
| J95-1 | 57522746 | 55694584 (96.82%) | 48.81 | 1518110 (2.64%) | 97.96 |
| J95-2 | 54936828 | 53552704 (97.48%) | 47.98 | 1093624 (1.99%) | 98.14 |
| J95-3 | 50112588 | 48795730 (97.37%) | 49.16 | 1041306 (2.08%) | 98.07 |
| J125-1 | 51435596 | 50328740 (97.85%) | 47.38 | 873746 (1.7%) | 98.26 |
| J125-2 | 56116664 | 54392104 (96.93%) | 49.77 | 1407218 (2.51%) | 97.93 |
| J125-3 | 48807236 | 47426834 (97.17%) | 48.26 | 1100572 (2.25%) | 98.10 |
| B65-1 | 47518824 | 45788238 (96.36%) | 48.01 | 1485002 (3.13%) | 97.75 |
| B65-2 | 40622062 | 39269470 (96.67%) | 50.06 | 1118380 (2.75%) | 97.77 |
| B65-3 | 45081614 | 43642502 (96.81%) | 48.27 | 1178906 (2.62%) | 97.85 |
| B95-1 | 58292330 | 56603702 (97.1%) | 49.11 | 1338248 (2.3%) | 98.05 |
| B95-2 | 50394128 | 49223592 (97.68%) | 48.76 | 902068 (1.79%) | 98.18 |
| B95-3 | 47285220 | 46032580 (97.35%) | 48.88 | 1005528 (2.13%) | 98.07 |
| B125-1 | 48935088 | 47639316 (97.35%) | 48.69 | 1022718 (2.09%) | 98.11 |
| B125-2 | 49062636 | 47699064 (97.22%) | 49.14 | 1075318 (2.19%) | 98.04 |
| B125-3 | 54076892 | 52576054 (97.22%) | 49.6 | 1171282 (2.17%) | 98.04 |

organic acid content between the two cultivars were observed in the over-ripening stage, we focused on DEGs between the two cultivars at 125 DAF. The KEGG pathway enrichment of these DEGs was analyzed. We found 9 DEGs in the enriched sugar and citrate metabolic pathways (Table 3 and Fig 3C).

## 3.4. Candidate genes associated with organic acid and sugar metabolism

Six DEGs related to malic acid metabolism were identified from the transcriptome profiles. The expression trends and their functional roles were detailed in Fig 4A. The expression levels of *PEPC2* in BL increased from 65 to 95 DAF and then decreased, whereas in JFZ it decreased from 65 to 95 DAF and then increased. The expression abundance of *cyNADP-ME2* decreased from 65 to 95 DAF, and then increased from 95 to 125 DAF. The expression levels of *VHA-A* and *VHP1* increased in JFZ and BL during fruit development. The expression level of *mNAD-MDH* in BL was significantly higher than in JFZ at 125 DAF (Fig 4B). In both JFZ and BL, the RPKM value of *cyNADP-ME2* was significantly higher in JFZ than in BL at 125 DAF

**Table 2. Summary of the *de novo* assembly.**

| | Number | Percentage (%) |
|---|---|---|
| Total assembled bases | 44034008 | - |
| Total unigenes | 45041 | - |
| GC | | 44.23 |
| Mean length (nt) | 977 | 2.17 |
| N50 length (nt) | 1555 | 3.45 |
| Unigenes (300–500 nt) | 10532 | 23.38 |
| Unigenes (500–1500 nt) | 16958 | 37.65 |
| Unigenes (1500–3000 nt) | 7574 | 16.82 |
| Unigenes (>3000 nt) | 1791 | 3.98 |

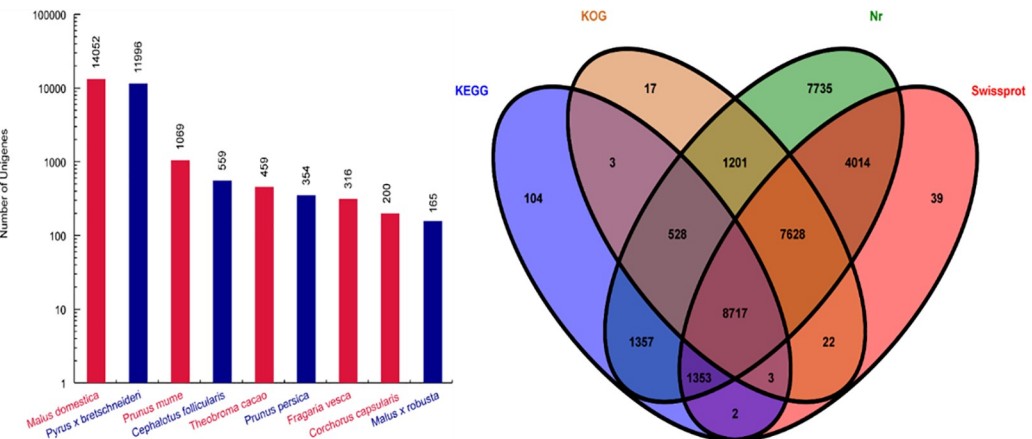

**Fig 2. Homology characteristics of E. japonica unigenes.** (A) Species distribution of Nr annotation; (B) Venn diagram of Nr, Swiss-Prot, KOG, and KEGG annotation.

(Fig 4B). The expression of *ALMT 9* and *cyNADP-ME2* was 2-fold lower in BL than in JFZ at 125 DAF (Fig 4B). The expression of *VHP1* observed in BL was 1.5-fold that of JFZ at 125 DAF (Fig 4B).

Our transcriptome studies identified six DEGs related to fructose and sucrose metabolism (Fig 5A). *NAD-SDH* transcripts increased with fruit development and were 2-fold higher in BL compared with JFZ at 125 DAF (Fig 5B). The expression level of *FRK1* showed the opposite trend and was significantly higher in JFZ compared with BL at 125 DAF (Fig 5B). *SPS1* was

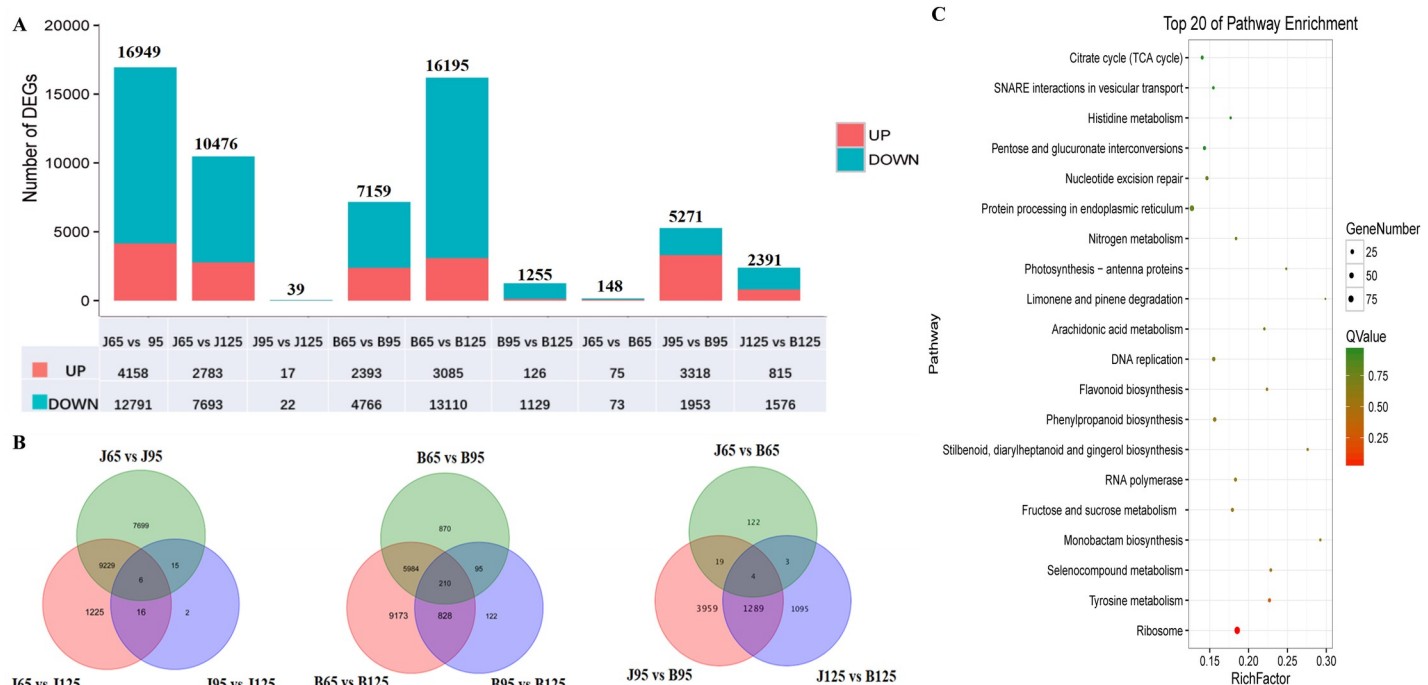

**Fig 3. Analysis of differentially expressed genes between JFZ and BL.** The number of differentially expressed genes between different samples (A). The Venn diagram showed the number of unigenes expressed in all three developmental stages of both cultivars (B). The top 20 enriched KEGG pathways were sorted by P-value for DEGs between the two cultivars at 125 DAF (C).

**Table 3. List of 12 DEGs associated with fructose, sucrose, and malic acid metabolisms at 125 DAF.**

| Unigene ID | Predicted Function | EC Numbers | Nr-ID | Source Organism | P-value |
|---|---|---|---|---|---|
| Unigene0026438 | Phosphoenolpyruvate carboxylase 2 | 4.1.1.31 | XP_009354255.1 | *Pyrus bretschneideri* | 0.001478 |
| Unigene0019317 | NAD-malate dehydrogenase | 1.1.1.299 | XP_009340262.1 | *Pyrus bretschneideri* | 0.001478 |
| Unigene0019373 | NADP-dependent malic enzyme 2 | 1.1.1.40 | XP_009367432.1 | *Pyrus bretschneideri* | 4.82E-07 |
| Unigene0022415 | Aluminum-activated malate transporter 9 | — | XP_008367579.1 | *M. domestica* | 0.000258 |
| Unigene0026223 | V-ATPase subunit A | — | AIZ49540.1 | *E. japonica* | 0.000396 |
| Unigene0025326 | Pyrophosphate-energized proton pump 1 | 3.6.1.1 | XP_008370385.1 | *M. domestica* | 3.51E-14 |
| Unigene0022361 | NAD-sorbitol dehydrogenase | 1.1.1.200 | XP_009371271.1 | *Pyrus bretschneideri* | 3.32E-19 |
| Unigene0020970 | Fructokinase | 2.7.1.4 | XP_008376805.1 | *M. domestica* | 0.003359 |
| Unigene0027245 | Sucrose-phosphate synthase 1 | 2.4.1.14 | XP_009358563.1 | *Pyrus bretschneideri* | 0.004309 |
| Unigene0021692 | Sucrose synthase in sucrose cleavage | 2.4.1.13 | XP_009348896.1 | *Pyrus bretschneideri* | 0.000491 |
| Unigene0024799 | Neutral invertase | 3.2.1.26 | XP_008339170.1 | *M. domestica* | 0.005409 |
| Unigene0011826 | Vacuolar invertase | 3.2.1.26 | AHK05786.1 | *E. japonica* | 4.39E-10 |

upregulated in response to fruit development in both JFZ and BL (Fig 5B). The expression level of *SS-C* increased in the two cultivars throughout fruit development and was significantly higher in BL compared with JFZ at 125 DAF (Fig 5B). *vAI* was downregulated in JFZ and upregulated in BL and was significantly lower in JFZ than in BL at 125 DAF (Fig 5B).

### 3.5. Verification of DEGs related to organic acid and sugar metabolism by quantitative reverse transcription PCR (RT-qPCR)

To further verify the accuracy of the transcriptome data, 12 enzyme-encoding genes associated with sugar metabolism and organic acids were selected for qRT-PCR analysis. The results obtained by qRT-PCR corroborated the transcriptomic data (Fig 6) indicating the robustness of the transcriptomic data.

### 3.6. Changes in organic acid and sugar-related enzymes during fruit development

The activity of PEPC was the highest at 95 DAF in both JFZ and BL and decreased thereafter; PEPC activity was higher in JFZ than in BL at 95 DAF (Fig 7A). In contrast, the activity of NADP-ME was minimum at 95 DAF and was higher in BL compared with JFZ at 125 DAF (Fig 7B). NAD-SDH activity increased during fruit development in both cultivars and was significantly higher in BL than in JFZ at 125 DAF (Fig 7C). SPS activity continuously increased with fruit development and had no significant differences between the two cultivars (Fig 7D). The two cultivars showed a similar pattern for mNAD-MDH activity (Fig 7E). Similar to SPS, AI acitivity also continuously increased with fruit development, but it was much higher in BL compared with JFZ at 125 DAF (Fig 7F). The activity of fructokinase (FRK) decreased with fruit development and was significantly higher in JFZ than in BL at 125 DAF (Fig 7G). The activity of SS-C was low at 65 DAF and 95 DAF but increased thereafter; SS-C activity was significantly lower in BL than in JFZ at 125 DAF (Fig 7H).

## 4. Discussion

Malic acid is known to be the most dominant organic acid in loquat fruits [14, 31, 32]. The present study showed that malic acid content increased as the fruit development progressed in the two loquat cultivars, JFZ and BL. These results corroborate previous studies that show increased malic acid accumulation in apple [33] and pineapple [34] during fruit development.

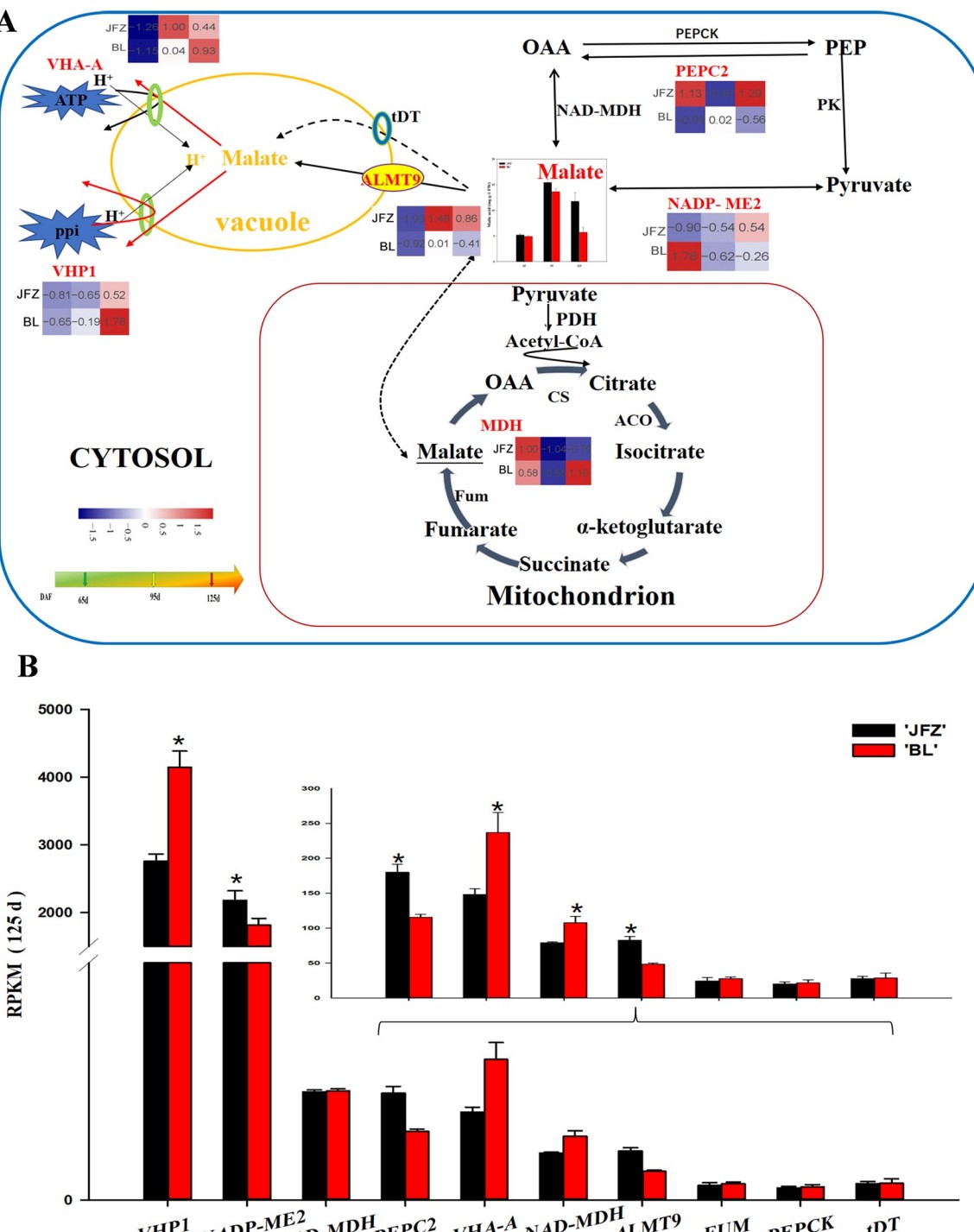

**Fig 4. Regulation of main organic acid (malate) metabolism during loquat fruit development.** (A) The transcript level trends were determined using log2-transformed values. The malic acid content is shown in columns. The color of the heat map from blue to red represents the gene transcript levels from low to high. (B) Transcript level analysis of genes involved in organic acid metabolism of two loquat cultivars at 125 DAF. Differences among the samples were analyzed by two (cultivars) × three (sampling dates) ANOVA. The asterisks represent $P < 0.05$. Each point represents the mean ± SD (n = 3). RPKM: reads per kilobase per million.

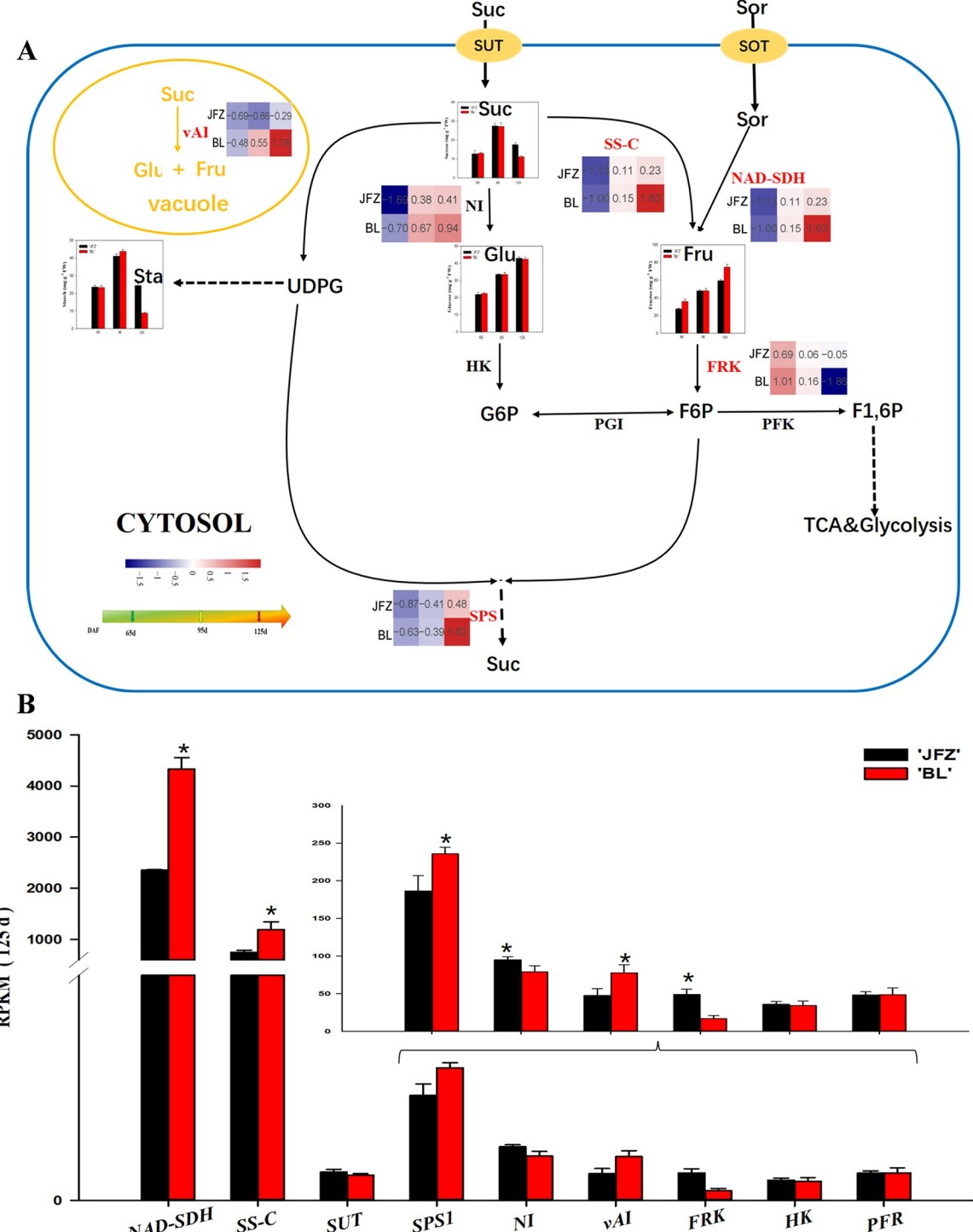

**Fig 5. Regulation of main sugar metabolism during loquat fruit development.** (A) The transcript level trends were determined using log2-transformed TMT values. The contents of fructose, sucrose, and glucose are shown in columns. The color of heat map from blue to red represents gene transcript levels from low to high. (B) Transcript level analysis of genes involved in sugar metabolism of two loquat cultivars at 125 DAF. Differences among the samples were analyzed by two (cultivars) × three (sampling dates) ANOVA. The asterisks represent P < 0.05. Each point represents the mean ± SD (n = 3). RPKM: reads per kilobase per million.

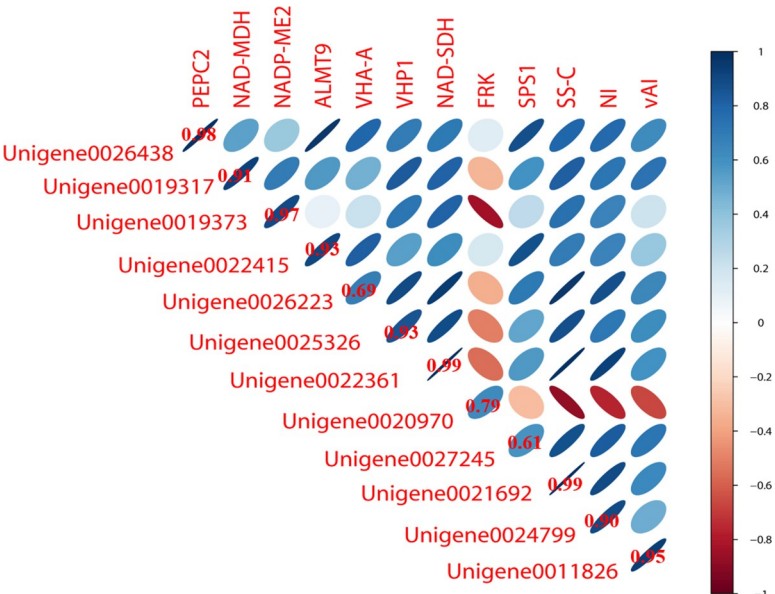

**Fig 6. Validation of digital gene expression using quantitative real-time PCR.** The stronger the correlation, the more flat the ellipse.

Our study showed that PEPC activity increased with increase in the malic acid content of the two cultivars suggesting a potential role in regulating malic acid synthesis. However, our study showed no difference in the content of malic acid between the two cultivars in the early stage of fruit development (from 65 to 95 DAF) as previously observed by Diakou et al. [35] and Lu et al. [36]. The accumulation of large amounts of malic acid during fruit development is attributed to its synthesis, transportation, and degradation. Chen et al. [31] showed that malic acid concentration in high-acid cultivars was significantly higher than in low-acid cultivars in the late stages of loquat fruit development. According to their study, the difference in fruit acidity of different genotypes is primarily due to the degradation and transportation of malic acid rather than its synthesis in the late stage of fruit development (from 95 to 125 DAF). Moreover, the cytosolic NADP-ME enzyme was found to play a crucial role in the degradation of malic acid. The activity of cyNADP-ME in 1-MCP-treated apples was higher than in untreated apples at 14 and 28 days of storage, indicating that 1-MCP-treated can promote the degradation of malic acid by reducing the activity of cyNADP-ME [37]. Our study showed that cyNADP-ME activity in the two cultivars was inversely related to the malic acid content during the entire fruit development period, and was significantly higher in BL than in JFZ at 125 DAF. This result indicated that the differences in cyNADP-ME activity may lead to the difference in malic acid content between high- and low-acid fruits. Similar results have been obtained for *Cerasus humilis* and peach [38, 39]. Moreover, BL showed higher transcript levels of *cyNADP-ME2* than JFZ at 125 DAF; however, a correlation between the expression level of *cyNADP-ME2* and its enzyme activity throughout fruit development was not observed. Therefore, we conclude that cyNADP-ME might be regulated by other genes or at the post-translational level during fruit development, which is consistent with the results observed by Yao et al. [40]. Aluminum-activated malate transporters (ALMTs) and the electrochemical gradient established by VHA and VHP are responsible for transporting malate from the cytosol into or out of the vacuole [41]. In grapes, the transient expression of the *VvALMT9*: GFP protein demonstrated that VvALMT9 facilitates the accumulation of malate in the vacuole of grape berries

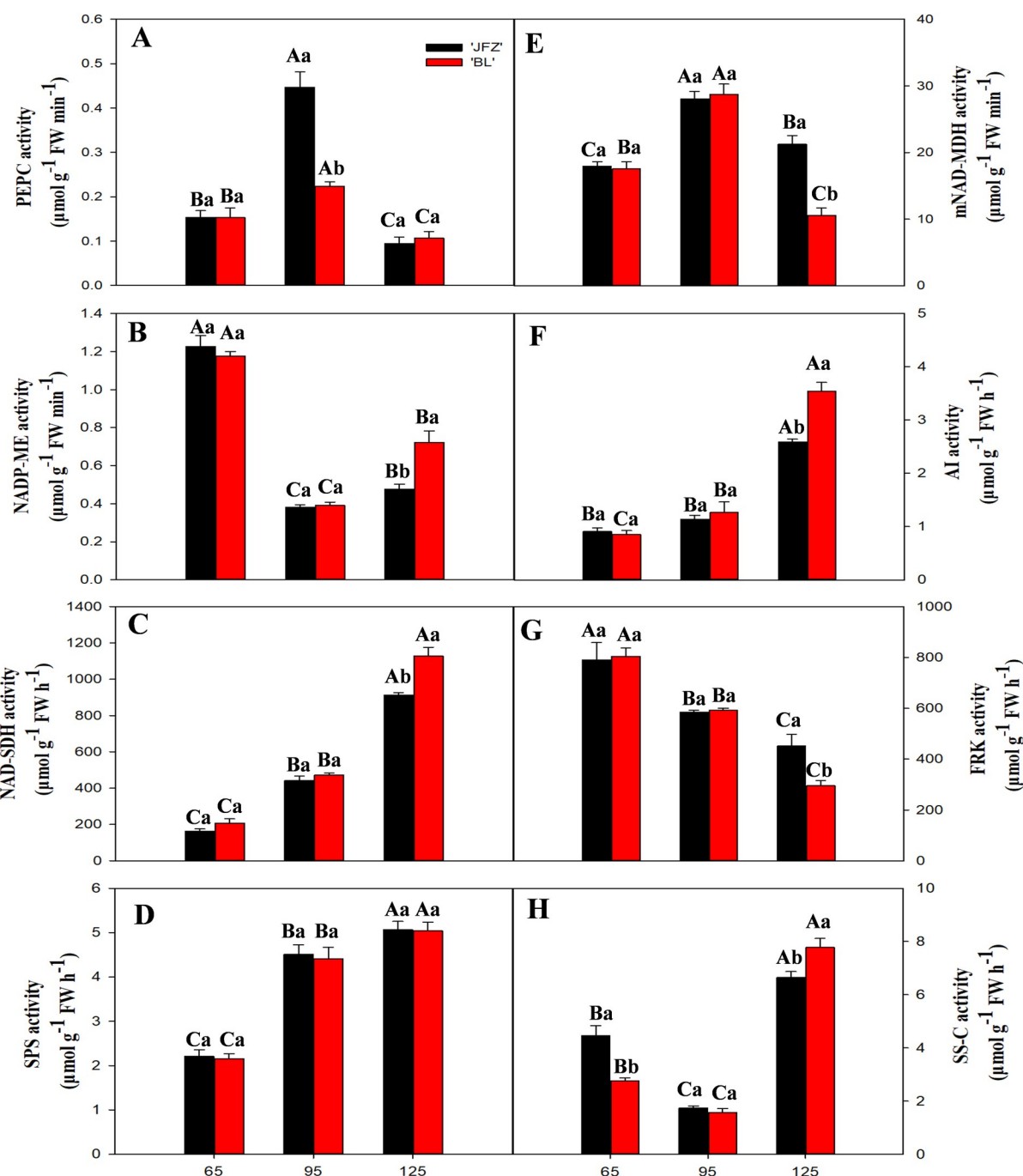

**Fig 7. Changes in activities of PEPC, NAD-MDH, NADP-ME, vAI, NAD-SDH, FRK, SPS, and SS-C during fruit of development and ripening.** The samples were analyzed for inter-cultivar differences and differences in the sampling dates for the two cutivars. Differences among the samples were analyzed by two (cultivars) × three (sampling dates) ANOVA. The different lowercase letters indicate significant differences between the two cultivars at the same time point at $p < 0.05$. The different capital letters indicate significant differences among different time points for the same cultivar at $p < 0.05$. Each point represents the mean ± SD (n = 3).

[42]. We observed that *ALMT9* in two cultivars was upregulated during fruit development, and its expression in JFZ was higher than in BL at 125 DAF. Therefore, ALMT9 may participate in the control of malic acid accumulation in the vacuole during fruit development. Similar

results have been obtained for tomato [43]. Transcriptomic analysis showed that the gene expression levels of *VHA-A* and *VHP1* increased in BL during fruit development, which is in contrary to the trend of increase in malic acid content from 95 to 125 DAF. These results indicated that the reduction of malic acid content in the later stage of fruit development may be attributed to the increased anion transport to compensate for the leakage from the tonoplast, which corroborated the results obtained by Terrier [44]. Our finding showed that the expression levels of *VHA-A* and *VHP1* in low-acid loquat fruits were significantly higher than in high-acid loquat fruits at 125 DAF. This may be attributed to the difference in tonoplast leakage, which caused differences in the malic acid content between the two cultivars. This result is consistent with the finding of Yao [45]. The gene expression level of *VHP1* in BL was 10-folder higher than *VHP1* in JFZ for most of the growth period, indicating that *VHP1* causes an increase in the activity of the proton pumps resulting in malate leakage across the tonoplast, which may be responsible for the slight decrease in malate concentration.

As with all other fruits of the family Rosaceae, fructose and sucrose are the primary sugars in loquat fruits. Our data clearly showed that the content of fructose in both cultivars was very low during the early stage of fruit development, and sharply increased in the late stage of fruit development, whereas the sucrose and starch content rapidly decreased in the later stages of fruit development. The same dynamic changes in the fructose and sucrose content were reported previously by Qin et al. [24]. The content of TSS was significantly higher in BL than in JFZ fruits at 125 DAF indicating that the difference in fruit sweetness between the cultivars is mainly due to the accumulation of soluble sugars in later stages of fruit development, which is consistent with previous observations in wolfberry varieties [46]. The NAD-SDH activity plays a vital role in converting sorbitol into fructose, which is supported by the higher NAD-SDH enzymatic activities and transcripts, and lower NAD-SDH levels in the low-sugar cultivar JFZ in later stages of fruit development. Li et al. [23] showed that the content of fructose is mainly determined by NAD-SDH enzymatic activities in apple fruits. Fructose is primarily phosphorylated by FRK. In apple, high FRK activity and low levels of fructose accumulation were observed in the early stage of fruit development [47]. Our study showed that there was a negative correlation between FRK activity and fructose content with the development of the fruit in the two cultivars. FRK is an important enzyme in the degradation of fructose [48]. The gene expression and activity of FRK in BL was lower than in JFZ at 125 DAF, which may have caused the differences in the fructose content in the two cultivars.

In addition to fructose, sucrose content is also abundant in mature loquat fruit. Many studies have suggested that sucrose accumulation mainly depends on the balance between sucrose degradation and synthesis, which is partially controlled by SPS, SUSY, and vAI [49, 50]. In many fruits, the content of sucrose increases with fruit development such as kiwifruit and apple [51, 52]. Interestingly, we found that the sucrose content decreased, while fructose rapidly increased, in the late stage of the development of loquat fruit. In citrus, this can be explained by the conversion of sucrose to hexose via sucrose degradation in the later stages of fruit development [53]. In our study, the content of sucrose in the two cultivars was negatively correlated with the activity and gene expression level of SS-C. Moreover, the activity and gene expression level of SS-C in the low-sugar cultivar JFZ was significantly higher than in the high-sugar cultivar BL at 125 DAF. Thus, it is reasonable that SS-C might cause a decrease in sucrose content in the late stages of fruit development, which is consistent with the results observed by Gao et al. [54]. Sucrose can also be converted to Glc and Fru by vAI in the vacuole. Itai et al. [25] demonstrated that treating Japanese pears with 1-MCP inhibits *vAI* gene expression, and prevents the loss of sucrose. In peach fruit, treatment with HA and jasmonic acid may reduce vAI activity, which in turn may increase the sucrose content during cold storage [55]. Notably, the accumulation of sucrose is controlled via SPS enzyme. Li et al. [56] reported that GB

treatment of peach fruit may significantly improve the sucrose content by enhancing the activities of SPS. A positive correlation between the SPS activity and *SPS1* gene expression in the two cultivars was found in our study. Also, SPS activity was negatively correlated with sucrose content in the early stage of fruit development of BL and JFZ. This suggests that SPS has a pivotal role in determining sucrose accumulation at the early stage of fruit development. Together, in these two cultivars, the synthesis of sucrose may be mainly regulated by SPS in the early stage of fruit development, whereas SS-C and vAI may be responsible for the degradation of sucrose in the late stages of fruit development.

## Supporting information

**S1 Fig.**
(TIF)

**S1 Table.**
(XLSX)

**S2 Table.**
(XLSX)

**S3 Table.**
(XLSX)

**S4 Table.**
(XLSX)

**S5 Table.**
(XLSX)

**S6 Table.**
(XLSX)

**S7 Table.**
(XLSX)

**S8 Table.**
(XLSX)

## Author Contributions

**Conceptualization:** Guo-Hua Zheng.

**Data curation:** Jing Zhang.

**Formal analysis:** Xue-Lian Zheng.

**Funding acquisition:** Xian-Qian Niu, Jin-Cheng Wu.

**Investigation:** Xu Chen.

**Resources:** Xue-Lian Zheng.

**Software:** Jing Zhang.

**Supervision:** Jin-Cheng Wu.

**Validation:** Jin-Cheng Wu.

**Visualization:** Xu Chen.

**Writing – original draft:** Jun Yang.

**Writing – review & editing:** Jun Yang.

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
