## [Decision Letter · Decision Letter 0]

17 Jun 2020

PONE-D-20-06341

Comparative Transcriptome Analysis Reveals Key Genes Potentially Related to Organic Acid and Sugar Accumulation in Loquat

PLOS ONE

Dear Dr. Zheng,

Thank you for submitting your manuscript to PLOS ONE. After careful consideration, we feel that it has merit but does not fully meet PLOS ONE’s publication criteria as it currently stands. Therefore, we invite you to submit a revised version of the manuscript that addresses the points raised during the review process.

We look forward to receiving your revised manuscript.

Kind regards,

Xiaoming Pang, PhD

Academic Editor

PLOS ONE

Additional Editor Comments:

Pay attention to the typos, for example, P43，polant families

Organic acids related taste was repeated several times in such a short introduction.

The rationale of present study and the use of the two cultivars was not well presented.

The discussion is better to be arranged under several subtitles or just combined with the results part.

Journal Requirements:

"NO"

"NO"

6. Please amend the manuscript submission data (via Edit Submission) to include author Jun Yang, Xue-Lian Zheng, Xu Chen.

7. Please amend your list of authors on the manuscript to ensure that each author is linked to an affiliation. Authors’ affiliations should reflect the institution where the work was done (if authors moved subsequently, you can also list the new affiliation stating “current affiliation:….” as necessary).

Reviewers' comments:

Reviewer's Responses to Questions

**Comments to the Author**

1. Is the manuscript technically sound, and do the data support the conclusions?

Reviewer #1: Yes

2. Has the statistical analysis been performed appropriately and rigorously? 

Reviewer #1: Yes

3. Have the authors made all data underlying the findings in their manuscript fully available?

Reviewer #1: Yes

4. Is the manuscript presented in an intelligible fashion and written in standard English?

Reviewer #1: Yes

5. Review Comments to the Author

Reviewer #1: The manuscript is well presented, and it is systematic analyses of major organic acids, sugar content, enzyme activities, and the expression of related genes during fruit development in two different loquat cultivars. It is well studied that the sugar content increased during fruit development, the organic acid content dramatically decreased in the later stages of fruit development. The DEGs in the later stages of fruit development between the two cultivars were analyzed in depth. The findings will help understand the molecular mechanism of organic acid and sugar formation in loquat, which will aid in breeding high-quality loquat cultivars.

Some comments:

-In the abstract part, the author only described the results and lacks summative sentences.

-There is a lack of detailed description of the experimental materials, why these two experimental materials were used, why these three sampling times were chosen?

-The genetic background of the two cultivars was somewhat different, so the credibility of DEGs was somewhat low.

-There are only three sampling time points, and if there were more time points, the results would be more reliable.

6. PLOS authors have the option to publish the peer review history of their article (what does this mean?). If published, this will include your full peer review and any attached files.

Reviewer #1: No

---

## [Author Response · Author response to Decision Letter 0]

30 Jul 2020

Professor Xiaoming Pang

Editor, Plos one

July 8, 2020

Dear Dr. Xiaoming Pang

Thank you so much for critiquing our submission titled "Comparative transcriptome analysis reveals key genes potentially related to organic acid and sugar accumulation in loquat." We reviewed the concerns from the reviewers and have improved our manuscript. We have also utilized the service of an English language editing company to improve the language and presentation of our manuscript. Therefore, we have incorporated the suggested changes and would like to resubmit our manuscript.

We have incorporated the changes suggested by the reviewers and have answered their questions to the best of our ability. Please see our point-to-point responses below.

Reviewer 1:

1. In the abstract part, the author only described the results and lacks summative sentences.

We have added the summative sentence of the article from 2- Page 2, line 35–43 in the abstract part.

2. There is a lack of detailed description of the experimental materials, why these two experimental materials were used, why these three sampling times were chosen?

We selected the cultivars ‘JiFangZhong’ and ‘BaiLi’ because ‘JFZ’ has high acid and low sugar, while ‘BL’ has low acid and high sugar. Both cultivars are late maturing, and have similar flowering patterns. In addition, we also provide pictures of the three periods of two varieties in S1 Fig. Therefore, the differences in sugar/acid ratios in the two cultivars would provide a good overview of their accumulation and metabolism and the related genes. Chen et al, Hirai M and Yang (reference 6, 9 and 12 in the revised manuscript) have showed that these three time points are the critical period during the accumulation of louquat fruit sugar and acid. Thus we chose these three developmental stages in this study.

3. The genetic background of the two cultivars was somewhat different, so the credibility of DEGs was somewhat low.

A previous study by Chen et al. (reference 8 in the revised manuscript) have indicated that the two cultivars in this study all belong to Eriobotrya japonica Lindl. Thus, We think the two cultivars have a consistent genetic background.We have added Eriobotrya japonica Lindl to the title of the article.

4. There are only three sampling time points, and if there were more time points, the results would be more reliable.

First, according to a large number of article (reference 6, 9 and 12 in the revised manuscript), the three periods selected in this paper are the three key periods for the accumulation of sugar and acid in loquat fruit. Then, Jiang et (in Supplementary literature) have showed that the flesh of pummelo cv. GXMY and its mutants HRMY and HJMY in three development periods of fruit were selected to investigate the candidate genes of carotenoid metabolism by the RNA-Seq technique. Besides, Zhang et (in Supplementary literature) have also showed that seeds of P. rockii and P. lutea at 20, 60 and 80 days were collected to study FA biosynthesis metabolism by the RNA-Seq technique. . Thus We think the three points taken in this article can explain the accumulation of sugar and acid mechanism during the development of loquat fruit. 

Format modification: 

We have modified the format strictly in accordance with the format requirements of Plos one.

Editor's Comments to Author: Pay attention to the typos, for example, P43，polant families

Organic acids related taste was repeated several times in such a short introduction.The rationale of present study and the use of the two cultivars was not well presented.

We have modified the polant typo to plant, and have carefully checked other typos in the article. We have deleted some sentences about Organic acids related taste in introduction, such as P55 and P63. We have added some content about the rationale of present study and the use of the two cultivars in S1 Fig, P13, P68-72 and P84-87.if there are any further corrections necessary, we will be happy to incorporate them.

We look forward to your response. Thanks again for helping us improve this submission for your esteemed journal.

Best regards,

---

## [Decision Letter · Decision Letter 1]

26 Aug 2020

Comparative Transcriptome Analysis Reveals Key Genes Potentially Related to Organic Acid and Sugar Accumulation in Loquat

PONE-D-20-06341R1

Dear Dr. Zheng,

We’re pleased to inform you that your manuscript has been judged scientifically suitable for publication and will be formally accepted for publication once it meets all outstanding technical requirements.

Kind regards,

Xiaoming Pang, PhD

Academic Editor

PLOS ONE

Additional Editor Comments (optional):

Reviewers' comments:

Reviewer's Responses to Questions

**Comments to the Author**

1. If the authors have adequately addressed your comments raised in a previous round of review and you feel that this manuscript is now acceptable for publication, you may indicate that here to bypass the “Comments to the Author” section, enter your conflict of interest statement in the “Confidential to Editor” section, and submit your "Accept" recommendation.

Reviewer #1: All comments have been addressed

2. Is the manuscript technically sound, and do the data support the conclusions?

Reviewer #1: Yes

3. Has the statistical analysis been performed appropriately and rigorously? 

Reviewer #1: Yes

4. Have the authors made all data underlying the findings in their manuscript fully available?

Reviewer #1: Yes

5. Is the manuscript presented in an intelligible fashion and written in standard English?

Reviewer #1: Yes

6. Review Comments to the Author

Reviewer #1: (No Response)

7. PLOS authors have the option to publish the peer review history of their article (what does this mean?). If published, this will include your full peer review and any attached files.

Reviewer #1: No

---

## [Editor Report · Acceptance letter]

12 Apr 2021

PONE-D-20-06341R1 

Comparative Transcriptome Analysis Reveals Key Genes Potentially Related to Organic Acid and Sugar Accumulation in Loquat 

Dear Dr. Zheng:

I'm pleased to inform you that your manuscript has been deemed suitable for publication in PLOS ONE. Congratulations! Your manuscript is now with our production department. 

Kind regards, 

on behalf of

Dr. Xiaoming Pang 

Academic Editor

PLOS ONE